# Cellulose/Zeolitic Imidazolate Framework (ZIF-8) Composites with Antibacterial Properties for the Management of Wound Infections

**DOI:** 10.3390/jfb14090472

**Published:** 2023-09-13

**Authors:** Valentina Di Matteo, Maria Francesca Di Filippo, Barbara Ballarin, Giovanna Angela Gentilomi, Francesca Bonvicini, Silvia Panzavolta, Maria Cristina Cassani

**Affiliations:** 1Department of Industrial Chemistry “Toso Montanari”, University of Bologna, Viale del Risorgimento 4, 40136 Bologna, Italy; valentina.dimatteo5@unibo.it (V.D.M.); barbara.ballarin@unibo.it (B.B.); 2Department of Chemistry “G. Ciamician”, University of Bologna, Via Selmi 2, 40126 Bologna, Italy; maria.difilippo5@unibo.it (M.F.D.F.); silvia.panzavolta@unibo.it (S.P.); 3Center for Industrial Research—Fonti Rinnovabili, Ambiente, Mare e Energia CIRI FRAME, University of Bologna, Viale del Risorgimento 2, 40136 Bologna, Italy; 4Center for Industrial Research—Advanced Applications in Mechanical Engineering and Materials Technology CIRI MAM, University of Bologna, Viale del Risorgimento 2, 40136 Bologna, Italy; 5Department of Pharmacy and Biotechnology, University of Bologna, Via Massarenti 9, 40138 Bologna, Italy; giovanna.gentilomi@unibo.it; 6Microbiology Unit, IRCCS Azienda Ospedaliero-Universitaria di Bologna, Via Massarenti 9, 40138 Bologna, Italy; 7Health Sciences and Technologies—Interdepartmental Center for Industrial Research (HST–ICIR), Alma Mater Studiorum—University of Bologna, Ozzano dell’Emilia, 40064 Bologna, Italy

**Keywords:** Zeolitic Imidazolate Framework-8, cellulose polymer, in situ single-step synthesis, bactericidal activity, chronic wound infections, cell compatibility

## Abstract

Metal–organic frameworks (MOFs) are a class of crystalline porous materials with outstanding physical and chemical properties that make them suitable candidates in many fields, such as catalysis, sensing, energy production, and drug delivery. By combining MOFs with polymeric substrates, advanced functional materials are devised with excellent potential for biomedical applications. In this research, Zeolitic Imidazolate Framework 8 (ZIF-8), a zinc-based MOF, was selected together with cellulose, an almost inexhaustible polymeric raw material produced by nature, to prepare cellulose/ZIF-8 composite flat sheets via an in-situ growing single-step method in aqueous media. The composite materials were characterized by several techniques (IR, XRD, SEM, TGA, ICP, and BET) and their antibacterial activity as well as their biocompatibility in a mammalian model system were investigated. The cellulose/ZIF-8 samples remarkably inhibited the growth of Gram-positive and Gram-negative reference strains, and, notably, they proved to be effective against clinical isolates of *Staphylococcus epidermidis* and *Pseudomonas aeruginosa* presenting different antibiotic resistance profiles. As these pathogens are of primary importance in skin diseases and in the delayed healing of wounds, and the cellulose/ZIF-8 composites met the requirements of biological safety, the herein materials reveal a great potential for use as gauze pads in the management of wound infections.

## 1. Introduction

Cellulose, an almost inexhaustible polymeric raw material produced by nature, has been widely used in various fields for thousands of years. It is a linear polysaccharide of α-D-glucose that assembles into nanofibers through multiple hydrogen bonding; the nanofibers are further organized to achieve a well-defined hierarchically fibrous structure with a large surface area [1,2]. Cellulose substances have many outstanding features such as biodegradability, renewability, flexibility, and high mechanical strength. In addition, the surface of natural cellulose nanofibers contains many hydroxyl groups, which makes it suitable for the introduction of functional molecules into heterogeneous reactions. Indeed, the structure of cellulose allows its surface to be chemically modified by processes such as oxidation, amination, and esterification without destroying its appealing intrinsic properties. In the biomedical field, cellulose fibers receive greater awareness than other biopolymers, and they outperform synthetics in many applications [3,4].

Cellulose-based biomaterials have tremendous potential for tissue healing applications because of their high absorptive capacity for exudates and moisture retention; recent biotechnological advances have made it possible to improve the properties of the material to cope with the growing impact of wound infections [5,6,7]. In fact, bacterial wound infections are a global healthcare issue that has been defined as a “silent epidemic” for their prevalence and profound effects on global health and patients’ lifestyles [8,9]. The severity of this burden is further worsened by the rise of antimicrobial resistance, which causes wound infections that are increasingly hard to treat with the current antibiotics. This calls for advanced wound dressing solutions combined with alternative antimicrobial therapeutics [10].

Hybrid materials made of cellulose and metal–organic frameworks (MOFs) conjugate the properties of both systems and have already demonstrated a great potential in several fields such as, to name just a few, air purification, water remediation, the catalytic degradation of pollutants, and biomedicine [11,12,13]. Our group has recently employed a simple, inexpensive, and easily scalable industrial paper process to prepare sheets of cellulose fibers coated with polyanilines. This conductive paper was used in the manufacturing of resistive and humidity touch sensors and as wearable disposable biomedical devices able to provide immediate diagnostic services such as heart rate or respiration activity monitoring [14,15,16]. In this framework, we chose Zeolitic Imidazolate Framework-8 (ZIF-8) as the active material for the functionalization of wood-based cellulose fibers via an in situ single-step method carried out at room temperature in aqueous media. ZIF-8, constructed from 2-methylimidazole and zinc ions, is a subclass of MOFs that are chemically stable in aqueous and basic media and have surface areas up to ca. 1800 m^2^g^−1^ [17,18,19,20]. These physical properties coupled with its excellent thermal stability and pH-responsive dissolution behavior (in acidic solutions) have motivated the investigation of ZIF-8 for biomedical applications, especially for drug delivery in cancer therapy [21,22,23,24] and for the controlled release of bioactive molecules such as antibacterial agents and metallic nanoparticles [1,2,25,26,27,28,29,30,31,32].

The intrinsically related antibacterial property of ZIF-8 crystals has been less explored [33,34], and there are no studies describing the effectiveness of cellulose/ZIF-8 hybrid materials on the pathogens responsible for human infections. Thus, an extensive characterization of the herein prepared functional materials was performed, and their potential as antibacterial agents was evaluated in vitro against reference bacterial strains and a panel of clinical isolates with different antibiotic resistance profiles. Furthermore, the cell compatibility and toxicity on non-malignant mammalian cells were ascertained in view of a possible application in the wound dressings.

## 2. Materials and Methods

### 2.1. Materials Synthesis

All chemicals were used as received from Sigma Aldrich (now Merck KGaA, Darmstadt, Germany); ultrapure water purified with the Milli-Q plus system (Millipore Co, resistivity over 18 MΩ cm, Burlington, VT, USA) was used in all experiments. Bare cellulose fibers (pine tree long fibers with sulphate treatment) were kindly provided by Cromatos S.r.l. (https://www.cromatos.com, accessed on 1 July 2023, Forlì, Italy).

### 2.2. Instrumentation

Powder X-ray Diffraction (PXRD) patterns were recorded in reflection mode by using a Philips X’Celerator diffractometer (PANalytical X’Pert PRO, Malvern PANalytical, Milano, Italy) equipped with a graphite monochromator. The 2θ range was from 4 to 45° with a step size of 0.100°, and a time per step of 120 s. CuKα (40 mA, 40 kV, 1.54 Å) was used. Morphological investigation was performed using a Leica/Cambridge Scanning Electron Microscope (SEM, Oxford Instruments, Abingdon, United Kingdom), and the images were obtained using a Leica/Cambridge Stereoscan 360 with INCA software. Digimizer software (version 5.8.0, MedCalc Software Ltd, Ostend, Belgium) was used to estimate the mean dimensions, averaging the measurements over at least 100 data points per sample. Thermogravimetric analysis (TGA) was performed using a Q600 SDT-TA instrument (New Castle, DE, USA). The analysis was performed under a nitrogen flow (100 mL/min) at a rate of 3.5 °C/min up to 120 °C and then 10 °C/min to 800 °C. The overall amount of zinc present in the different samples was determined by means of Agilent 4210 Molecular Plasma-Atomic Emission Spectroscopy (MP-AES, AgilentSanta Clara, CA, USA). Zinc lines at 472.215 and 481.053 nm were used. The analyses were conducted by comparison with five calibration standards (2, 20, 30, 50, 100 ppm), prepared with dilution to 100 mL of a 1000 ppm zinc standard (Carlo Erba Reagents, Cornaredo, Italy). The cellulose/ZIF-8 hybrid samples were analyzed after first digesting the solids (ca. 10 mg) with small amounts of aqua regia and then diluting with 1.0 M nitric acid (Normatom®, VWR International, Milan, Italy) in 25 mL flasks, while ZIF-8 alone dissolves completely in 1.0 M nitric acid at room temperature. Results from this analysis represent the mean value of three different determinations. The adsorption isotherm was measured using a static volumetric apparatus (ASAP 2020 Surface Area and Pore Size Analyzer, Micromeritics, Norcross, GA, USA). The samples were degassed at 1 × 10^−3^ mbar at 100 °C (ramp rate of 10 °C/min) for 2 h prior to measurement; the elaboration of data was carried out with the MicroActive Software (version 4, Micromeritics Instrument Corporation, Norcross, GA, USA); Langmuir and Brunauer-Emmett-Teller (BET) isotherms models were used for the estimation of the specific surface area according to the reference [35,36]. The size and distribution of pores was estimated by using the Density Functional Theory (DFT). Attenuated total reflectance-Fourier transform infrared (ATR-FTIR) analyses were performed with a Perkin Elmer Spectrum Two spectrophotometer, equipped with a Universal ATR accessory (Waltham, MA, USA) in the range 4000–400 cm^−1^ with a resolution of 0.5 cm^−1^.

### 2.3. Synthesis of Cell@ZIF-8 and Cell2@ZIF-8

To prepare cellulose with adsorbed ZIF-8, the following steps were carried out: i) 2.5 g of cellulose fibers was thoroughly dispersed in 90 mL of distilled water until a pulp suspension was obtained and ii) 2.30 g of Zn(OAc)_2_∙2H_2_O (10.48 mmol, MW: 219.51 g/mol) was dissolved in 10 mL of water and added to the cellulose suspension. The suspension was stirred for 1 h so that Zn^2+^ ions were adsorbed on the cellulose. Successively, a solution containing 8.60 g of 2-methylimidazole (2-HmIM, 104.8 mmol, MW: 82.20 g/mol) dissolved in 38 mL of water was added. The stoichiometric ratio of Zn^2+^:2-HmIM was 1:10. After a few minutes, a milky suspension was obtained with a pH at around 8–9, measured with litmus paper. The suspension was stirred for 24 h at room temperature and then shaped using a wooden sieve (52 × 74 mm size, A8 paper format, Appendix A) and thoroughly washed with water until the washings reach neutrality. The sheets were first dried in air at room temperature and then pressed at 50 bar (P50 AXA manual hydraulic press, Paris, France) for 60 s, obtaining a mean thickness of 0.2 mm. The thermal activation of the *as-synthesized* Cell@ZIF-8 was carried out at 100 °C and 10^−2^ mbar until there were no more imidazole sublimates (typically after ca. 3 h). The sample Cell2@ZIF-8, prepared with a double quantity of cellulose, was shaped using a wooden sieve with double the surface area (105 × 148 mm, A6 paper format) to maintain the same mean thickness. All the materials were stored in a desiccator.

### 2.4. Synthesis of ZIF-8

For comparison, a sample of ZIF-8 was prepared following a method in the literature [37]: 5 mL of an aqueous solution of Zn(OAc)_2_·2H_2_O (1.37 mmol, 0.27 M) was rapidly added to a solution of 2-methylimidazole (5 mL, 13.64 mmol, 2.7 M). The resulting suspension (Zn^2+^:2-HmIM = 1:10) was stirred (400 rpm) at room temperature for 24 h. Then, it was centrifuged and washed with CO_2_-free water until the washings had a pH of ca. 7 (litmus paper) before air drying in an oven at 70 °C for 24 h. After thermal activation at 100 °C and 10^−2^ mbar, 228 mg of white powder was obtained (yield: 73% based on zinc). Zn analysis: 27.2 ± 0.4% (calculated for Zn(mIM)_2_: Zn, 28.7%). The *activated* material was stored in a desiccator.

### 2.5. Bacterial Strains

The in vitro antibacterial activity of the cellulose/ZIF-8 composites (*as-synthetized*/*activated* Cell@ZIF-8 and Cell2@ZIF-8) was assayed against a panel of reference Gram-positive and Gram-negative strains, including *Staphylococcus aureus* (ATCC 25923), *Staphylococcus epidermidis* (ATCC 12228), *Escherichia coli* (ATCC 25922), and *Pseudomonas aeruginosa* (ATCC 27853). These reference strains were obtained from the American Type Culture Collection. In addition, the effectiveness of the *activated* Cell@ZIF-8 sample was evaluated by testing 8 clinical isolates of *S. epidermidis* and *P. aeruginosa*, identified using standard procedures and confirmed using the MALDI Biotyper System using matrix-assisted laser desorption ionization time of flight mass spectrometry (MALDI-TOF MS, Bruker Daltonik, GmbH, Oelde, Germany). These strains were profiled for their antibiotic susceptibility using the Vitek2 semiautomated system (bioMerieux, Lyon, France) and according to EUCAST guidelines [38].

All bacterial strains were routinely grown on a 5% blood agar plate (Biolife Italiana s.r.l., Milan, Italy) at 37 °C, and 24 h cultures were used for the experiments.

### 2.6. In Vitro Assessment of Cellulose/ZIF-8 Antibacterial Activity

The antibacterial properties of the cellulose/ZIF-8 composites, both *as-synthetized* and *activated*, were evaluated by testing cylindrical disk-shaped samples (6 mm in diameter) with a standardized disk-diffusion test in compliance with the International guidance documents [38,39]. Briefly, the surface of Mueller–Hinton agar plates (MHA) (Biolife Italiana S.r.l., Milan, Italy) were inoculated with the bacterial suspensions prepared in PBS (pH 7.2) and adjusted to an Optical Density at 630 nm (OD_630nm_) of 0.08–0.1, corresponding to 10^8^ CFU (colony forming units)/mL. Sample disks, together with bare cellulose disk-shaped controls, sterile disk controls (Liofilchem S.r.l., Teramo, Italy), and antibiotic-loaded positive controls (GNT, gentamicin; VNC, vancomycin, Liofilchem S.r.l., Teramo, Italy), were placed on the agar plates, and after 24 h of incubation at 37 °C, the diameter of the inhibition zone (corresponding to the bacterial-free zone around the disks) was measured with a ruler to the nearest whole millimeter. Disk-shaped samples obtained from different syntheses and preparations were tested to verify the repeatability of the procedures.

In addition, ZIF-8 powders prepared following the previously described protocols, *as-synthetized*, and thermally activated, were resuspended in PBS (9% *w*/*w*), and the resulting samples were tested by means of an agar cup diffusion method carried out on a MHA plate. In detail, the bacterial inoculum was spread on the solid medium, as previously described, and, thereafter, a cork borer was used for making wells on the agar plate (6 mm in diameter). Each well was subsequently filled with 100 µL of test sample and with a 10-fold dilution. The plates were incubated for 24 h at 37 °C, and zones of inhibition were measured.

### 2.7. Cells

Human lung fibroblasts (HEL 299; ATCC CCL137) were used as a model system to investigate the overall effect of the herein synthetized materials on mammalian cells. The cell line was obtained from the American Type Culture Collection and routinely cultured in Eagle's Minimum Essential Medium (EMEM, Gibco, Life technologies, Milan, Italy), supplied with 10% FBS (Fetal Bovine Serum, Carlo Erba Reagents, Milan, Italy), 100 U/mL penicillin, and 100 µg/mL streptomycin) at 37 °C and 5% CO_2_.

### 2.8. Cell Viability and Cytotoxicity

The safety profile of the samples was evaluated by measuring the cell metabolic activity of living cells and the activity of the lactate dehydrogenase (LDH) enzyme released in the cell medium from death cells. For the bioassays, cells were seeded onto a 12-well tissue culture plate at a concentration of 7 × 10^4^ cells/well. Following 24 h of incubation, the 12-well cell culture inserts with a pore size of 0.4 µm PET (polyethylene terephthalate) permeable membrane were used to expose cell monolayers to the disk-shaped samples (Figure 1).

After 48 h of incubation, the medium was recovered from the two compartments of the Transwell system (apical and basolaterial sides), centrifuged to remove potentially interfering molecules and cell debris, and assayed with the Cytotoxicity LDH Kit-WST (Dojindo Molecular Technologies, Rockville, MD, USA) according to manufacturer’s instructions. The cell viability and proliferation of cell monolayers were assessed by using the alamarBlue™ HS Cell Viability Reagent (Invitrogen, Thermo Fisher Scientific Inc. Waltham, MA, USA). LDH activity and cell viability were expressed as percentage values relative to both untreated cell controls and lysed reference controls.

In addition, Hel 299 monolayers were exposed to some selected samples for 24 h in the double-chambered system, then the inserts with the cellulose fibers/ZIF-8 composites were removed, and cells were further incubated in fresh medium for 48 h to measure the cell survival percentage in response to the treatment by adding the alamarBlue™ HS Reagent. Cell monolayers were also stained with crystal violet (0.1% in water) to image morphological changes due to the biomaterials.

## 3. Results and Discussion

### 3.1. Synthesis and Characterization of Cellulose/ZIF-8 Composites

In order to find the optimal conditions, several preparations were made while varying the mass of cellulose and the quantities and concentrations of Zn(OAc)_2_∙2H_2_O and 2-HmIM, although the molar ratio Zn/2-HmIM was always maintained at 1:10. Such a parameter was chosen after a careful evaluation of what was reported in the literature regarding the synthesis of ZIF-8 in water, for which the efficient formation of highly crystalline *sodalite*-type ZIF-8 requires a high concentration and a large excess of the ligand compared to the synthesis carried out in organic solvents [37,40]. As a consequence, to investigate the effect of the different concentrations of ZIF-8 in cellulose sheets, a synthesis was made, doubling the amount of cellulose and the area of the wooden sieve. Only the conditions reported in Table 1 led to the formation of workable cellulose/ZIF-8 flat sheets containing *sodalite*-ZIF-8 as the only crystalline phase (the whole preparation process is reported in Appendix A).

The actual ZIF-8 wt% loading in the cellulose/ZIF-8 composites was calculated by comparing the Zn wt% found by ICP analyses with the theoretical Zn wt% loading values. After the synthesis, the *as-synthesized* samples were thermally activated at 100 °C and 10^−2^ mbar to completely remove the excess of 2-HmIM still present inside the pores.

The SEM images (Figure 2) showed the characteristic rhombic dodecahedron morphology of *sodalite*-ZIF-8 and confirmed the presence of well-dispersed ZIF-8 crystals adsorbed on the surface of cellulose with a uniform size distribution and a dimensional range of 1.0−1.2 μm [41,42,43]. The average size of ZIF-8 crystals on the *activated* Cell@ZIF-8 sample was lower than that obtained for the Cell2@ZIF8 sample (1.2−1.5 μm, Appendix A). This size difference is in keeping with previous observations and attributed to the reduced amount of free Zn^2+^ available for the nucleation of ZIF-8 due to the concomitant interactions of Zn^2+^ with the hydroxy groups present on the cellulose fibers [44]. The SEM images and Inductively Coupled Plasma (ICP) analysis confirmed that to keep the Zn/2-HmIM 1:10 ratio but have a lower loading on cellulose, it is sufficient to vary the amount of cellulose.

The ATR-FTIR spectra of *as-synthetized* and *activated* Cell@ZIF-8, bare cellulose, and pure ZIF-8 are compared in Figure 3 (for Cell2@ZIF-8 see Appendix A). The characteristic bands of cellulose fibers, including -OH stretching and bending vibrations at 3334 and 1636 cm^−1^, the C-O vibration at 1300–900 cm^−1^, and the C-H vibration at 1500–1300 cm^−1^, were detected [14]. The intense band centered at 1037 cm^−1^ corresponds to the vibrations of C-O-C in the entire pyranose ring. These peaks were also observed in the cellulose/ZIF-8 composites together with those attributed to ZIF-8: the band at 1584 cm^−1^ for the C=N stretching mode in the imidazole ring; the bands at 1500–600 cm^−1^ assigned to the entire imidazole ring stretching or bending; and the diagnostic peak at 421 cm^−1^ ascribed to Zn-N stretching vibrations [45]. The relative intensities of the bands at 1037 and 421 cm^−1^ changed in the two composites (Appendix A) in agreement with the different ZIF-8 wt% content.

The TGA curves, carried out under nitrogen on the *as-synthetized* and *activated* Cell@ZIF-8 samples, are shown in Figure 4 together with the TGA curve of bare cellulose.

Bare cellulose showed two weight losses, one below 100 °C (adsorbed humidity) and the other at the onset temperature of about 330 °C, which corresponds to the pyrolysis of cellulose. In addition to the previous ones, the *as-synthetized* Cell@ZIF-8 sample had two other weight losses: one at the onset temperature of 160 °C, corresponding to the loss of unreacted imidazole and structural water, and one at around 580 °C, corresponding to the degradation of ZIF-8 [29,46]. The sample showed a residual of 23% at about 790 °C, which is higher than that obtained for the bare cellulose, which is equal to 11%. The thermogram of the *activated* Cell@ZIF-8 shows a weight loss due to an unreacted imidazole lower than that of the *as-synthesized* sample, in tune with the activation treatment performed on the sample. The Cell2@ZIF-8 sample shows lower losses due to the higher amount of cellulose with respect to the ZIF-8 (Appendix A).

The XRD patterns of *activated* Cell@ZIF-8, bare cellulose, and *activated* ZIF-8 samples are shown in Figure 5. The peaks of ZIF-8 at 2θ = 7.3°, 10.4°, 12.7°, 14.7°, 16.4°, and 18.0° associated with the crystal planes (110), (220), (211), (220), (310), and (222) of *sodalite*-ZIF-8 crystals are clearly recognizable in the diffractograms of cellulose/ZIF-8 composites before and after the activation procedure [47,48]. Even the XRD diffractogram of *activated* Cell2@ZIF-8, reported in Appendix A, confirmed the presence of a single-phase *sodalite* topology. These results are in agreement with the studies by James et al. [46] that claimed that *sodalite* ZIF-8 is in a stable phase up to 200 °C. Different intensities of the diffraction peaks can be ascribed to a different orientation of the crystals on the cellulose paper.

The surface area of both *activated* Cell@ZIF-8 and Cell2@ZIF-8 has been analyzed. In Table 2, both the BET and Langmuir surface areas are reported, even though the isotherms of both samples (Appendix A) showed a typical Langmuir behavior. As can be seen, the surface area of Cell@ZIF-8 is higher than that of the Cell2@ZIF-8 sample due to the higher ZIF-8 content; notably, if the specific surface area is calculated using only the mass of ZIF-8 present in the composite, the specific surface area of both samples would be in the typical range between 1500 and 1700 m^2^/g, as expected for these materials [19]. The size distribution of pores was obtained using the DFT (Appendix A), and 100% of the surface area was due to microporosity (<2 nm) in both samples. These results, together with the total pore volumes, are in agreement with those found in the literature [49]. The surface area of the pristine cellulose was below the minimum sensibility of the instrument, with a negligible contribution (<<1 m^2^/g, isotherm not reported) to the whole surface area of the samples.

### 3.2. Microbiological Results

In a preliminary set of experiments, bare cellulose fibers as well as cellulose/ZIF-8 composites were assayed against four bacterial reference strains. The results indicated that the bare cellulose sample did not interfere with bacterial proliferation, while cellulose samples containing ZIF-8 crystals displayed antibacterial activity towards all the tested strains but with a different degree of potency, as demonstrated by the diameters of the bacterial inhibition (Appendix A). No variations in the performance of the different batches were measured when testing samples for up to 1 year from their synthesis, indicating the stability of the cellulose-based materials. All these findings highlighted the suitability of the wood-based cellulose as an excellent platform for the anchoring of the ZIF-8 crystals and suggested the ability of the ZIF-8 based composites to act as a reservoir for metal ions.

The antimicrobial effect of MOFs is well described in the literature and is related to the dissolution of Zn^2+^ in the surrounding environment, leading to the rupture of the bacterial cell wall and the destruction of the cell membrane, thus causing the leakage of the cytoplasm. In addition, zinc ions interfere with intracellular biochemical pathways such as the generation of reactive oxygen species and the cell cycle mechanism, resulting in bacterial death [37,50]. Having a multi-faced mechanism of activity, MOFs have a strong and rapid killing effect on a broad spectrum of microorganisms, including intracellular pathogens [28].

Further investigations were carried out on *S. epidermidis* and *P. aeruginosa* species, two common pathogens potentially involved in the pathogenesis of skin diseases and wound infections [51,52] against which the cellulose/ZIF-8 composite flat sheets in the form of gauze pads could be of clinical relevance [10].

Table 3 displays the microbiological results of the different samples assayed against the Gram-positive and Gram-negative reference strains. Both samples, the *as*-*synthetized* and *activated* cellulose fibers/ZIF-8 crystals, exhibited remarkable inhibition regardless of the tested strains and without significant differences. Considering that the *as-synthetized* samples contained a high amount of unreacted 2-HmIM (up to 20%), it is possible to speculate that the organic part in the framework structure did not contribute to increasing the antibacterial effects of the ZIF-8 crystals. On the other hand, comparing the data obtained with the *activated* Cell@ZIF-8 and Cell2@ZIF-8 samples, some slight differences appeared, as demonstrated in Figure 6 and Appendix A. Samples prepared with twice the amount of cellulose, thus having a reduced content of ZIF-8 per area, were able to stop bacterial proliferation, but a higher variability in the diameters of inhibition was measured, possibly due to a less controlled release and diffusion of the ZIF-8 crystals from the cellulose fibers. For this reason, the *activated* Cell@ZIF-8 sample proved to be a promising candidate as hybrid antibacterial material.

The antibacterial property of the ZIF-8 crystals was finally examined by applying the agar cup diffusion method. The ZIF-8 samples proved to have excellent antimicrobial effect (Appendix A) regardless of the activation procedure and a concentration-dependent inhibitory activity was measured, indicating that the bactericidal effect of ZIF-8 is due to the dissolution of zinc ions in the medium instead of the released and associated pH increase of 2-HmIM [34].

### 3.3. Cell Compatibility

The existing literature on the biocompatibility of ZIF-8 and other MOF materials consists of investigations limited to a one-dimensional aspect such as cell as cell viability, mainly in cancer cell lines [37]. Herein, a non-malignant mammalian cell line (HEL 299 fibroblasts) was used as a model system to establish the feasibility of the material for biomedical applications by using different assays. In particular, the effects of the materials on cell metabolism (i.e., cell viability), together with the activity of the lactate dehydrogenase released from damaged cells (i.e., cytotoxicity), were quantitatively evaluated. This experimental outline, which combines two measurements, is desirable since the sole assessment of the cell metabolic activity does not allow for conclusions on the cytotoxicity and thus on the adverse effects of the biomaterials on cells and tissues.

In this study, the cellular response to the Cell@ZIF-8 composites was tested in double-chambered 12-well plates. Each well has a permeable PET-membrane suitable for drug transport studies [53] that separates the cell monolayer (in the lower chamber) from direct contact with the disk-shaped sample of cellulose fibers (in the upper chamber) but allows for maximum diffusion through the membrane pores. Thus, the Transwell system was used to evaluate the impact of the sustained release of ZIF-8 on mammalian cells.

As a first step in the in vitro evaluation of the materials, samples were assayed on Hel 299 cells, and cell viability and cytotoxicity were measured and compared to the cell response of an untreated control incubated with a sterile paper disk [Table 4]. The results indicated that the bare cellulose sample did not affect at all the cell metabolism of Hel 299 cells, confirming the biocompatibility of the material. Considering the ZIF-8-functionalized samples, both *as-synthetized* composites displayed strong cytotoxicity. When Hel 299 cells were incubated with *as-synthetized* Cell@ZIF-8 and Cell2@ZIF-8 samples, cell viability was completely nullified, and LDH percentage values were higher than the acceptable levels of biocompatibility for medical device materials (<30%) [49]. Remarkably, in our experimental conditions, both Cell@ZIF-8 samples tested after the thermal activation procedure exhibited good biocompatibility with a moderate reduction in cell viability but negligible levels of LDH, which is a direct measure of cell death. Considering that the activation removes the guest molecules of 2-HmIM trapped in the pores of the nanocrystal structure, it is possible to speculate that the unreacted organic ligand is mainly responsible for the cytotoxicity and that thermal activation is necessary to devise Cell@ZIF-8 composites with a safety profile.

Having demonstrated the excellent antibacterial potential of the *activated* Cell@ZIF-8 sample and its cell compatibility, a different experimental outline was designed on mammalian fibroblasts to measure the cell survival rate of Hel 299 upon a 24 h treatment. For this purpose, the cell monolayer was treated with the *activated* Cell@ZIF-8 sample for 24 h; thereafter, the disk-shaped biomaterial was removed from the apical chamber, and the cells were incubated for a further 48 h in a fresh medium. Interestingly, Hel 299 viability was completely restored in terms of metabolic activity, as no differences were measured compared to the untreated control (cells incubated with the sterile paper disk); in addition, the treated cells presented with a regular monolayer fibroblast-like morphology, as detailed in Figure 7. Overall, the biological assays demonstrated the strong and selective antibacterial efficiency of the *activated* Cell@ZIF-8 sample on Gram-positive and Gram-negative strains.

### 3.4. Clinical Isolates

The *activated* Cell@ZIF-8 sample selectively inhibiting bacterial growth was also assayed against a panel of clinical isolates of *S. epidermidis* and *P. aeruginosa* presenting different antimicrobial resistance to many classes of antibiotics. The data are reported in Table 5. Remarkably, the composite material proved to be active against all the tested isolates, regardless of their antibiotic resistance profile. This is clinically relevant considering that isolates may present phenotypic and genetic heterogeneity compared to laboratory reference strains, thus some diversities in susceptibility may occur. Bacterial strains that resistant to one antibiotic are frequently observed at the wound site (88.2% of isolates), and multi-drug resistance (MDR) strains are increasingly being identified (29.2% of isolates) [52]. 

Given that the antibacterial action of ZIF-8 is associated with physical damage to bacterial cells (rather than with a particular metabolic process, as seen with traditional antibiotics), the herein prepared material is a promising support of, or alternative to, antibiotics to control MDR bacterial infections, possibly preventing drug resistance development.

## 4. Conclusions

In the present study, natural cellulose fibers were functionalized with ZIF-8 crystals, and an effective antibacterial material was produced. The ZIF-8 was prepared by in situ growth on the cellulose fiber surfaces in a purely aqueous system at room temperature; hence, the synthesis protocol used in this study is in line with the principles of green chemistry, and thanks to the favorable working conditions, it offers promising prospects for production on an industrial scale. The *activated* Cell@ZIF-8 sample can be considered a suitable wound dressing material, for example, in gauze pad form, to prevent and treat bacterial wound infections, including those related to MDR pathogens. In fact, it demonstrated potent bactericidal activity against Gram-positive and Gram-negative strains without adversely affecting the Hel 299 cell line, used as an experimental model system of mammalian cells. Of note, the composite material had a bactericidal effect against all the tested clinical samples collected from biological specimens, regardless of their antibiotic susceptibility patterns, suggesting the suitability of the reagent in the management of infections with MDR bacteria. The good biocompatibility of the Cell@ZIF-8 sample broadens the application fields of the material and provides the possibility to explore ZIF-8-based materials as drug delivery systems for many types of bioactive molecules.

## Figures and Tables

**Figure 1 jfb-14-00472-f001:**
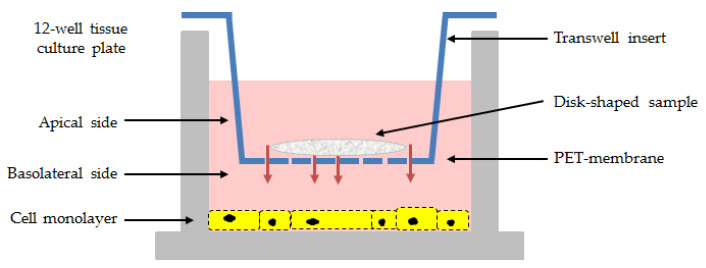
Schematic representation of the in vitro test for the cell compatibility assessment of *as-synthetized* and *activated* samples.

**Figure 2 jfb-14-00472-f002:**
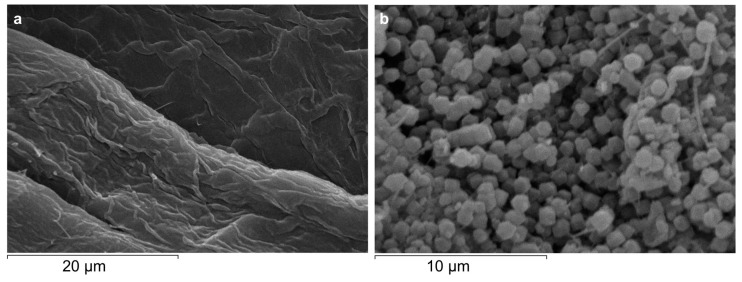
SEM images of bare cellulose (**a**) and *activated* Cell@ZIF-8 (**b**) at different scales (20 and 10 µm).

**Figure 3 jfb-14-00472-f003:**
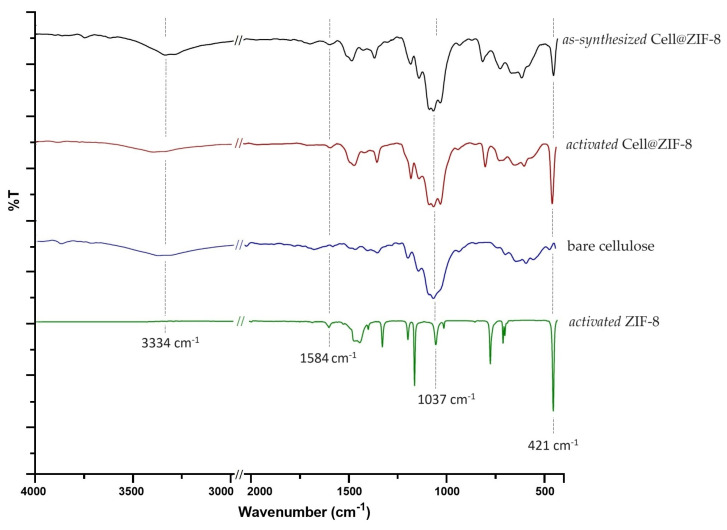
Comparison between the IR spectra of bare cellulose, *as-synthetized* and *activated* Cell@ZIF-8, and *activated* ZIF-8 samples.

**Figure 4 jfb-14-00472-f004:**
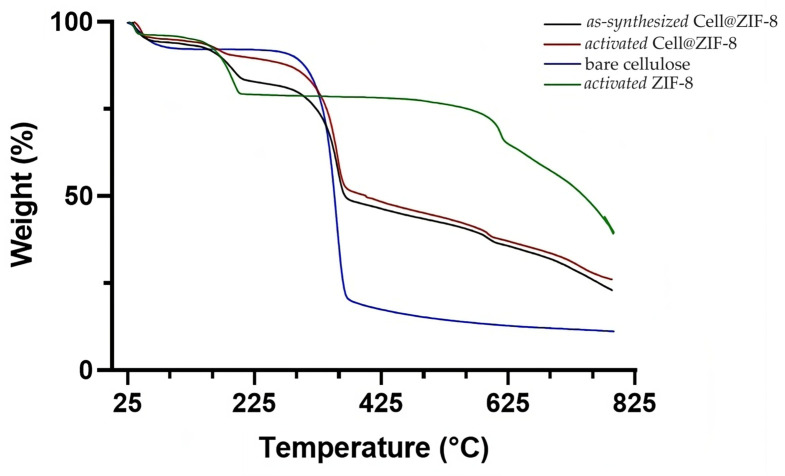
Comparison between the thermograms of bare cellulose (blue), Cell@ZIF-8 (black: *as-synthetized*, red: *activated*), and ZIF-8 (green) samples.

**Figure 5 jfb-14-00472-f005:**
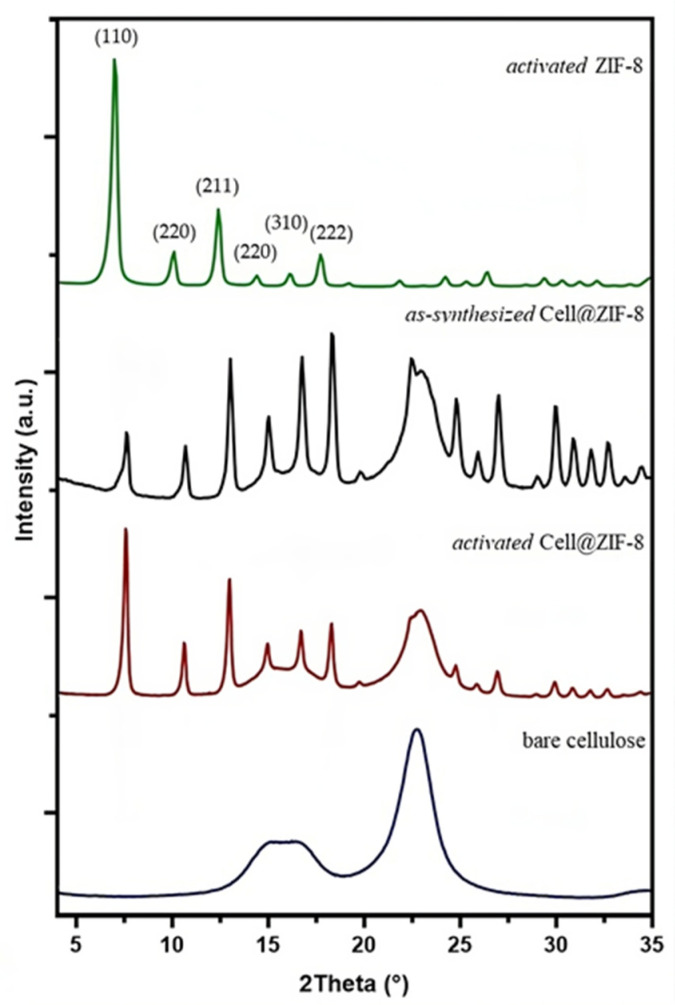
Comparison between the XRD diffractograms of *as-synthetized* and *activated* Cell@ZIF-8, bare cellulose, and *activated* ZIF-8 samples.

**Figure 6 jfb-14-00472-f006:**
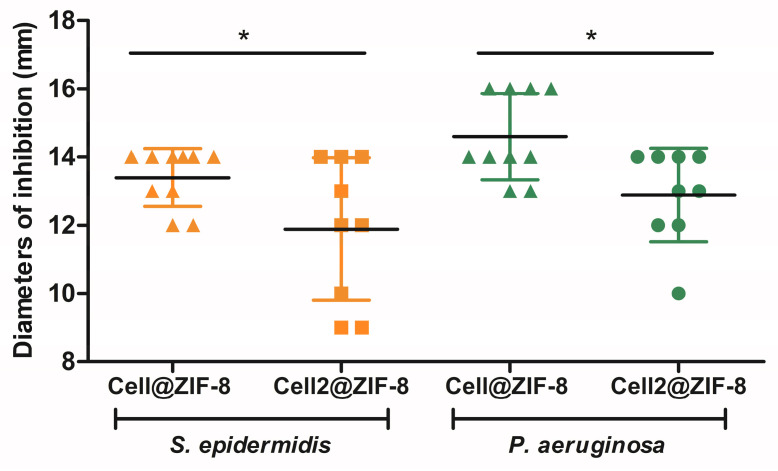
Scatter dot plot of the diameters of inhibition (in millimeter) of the *activated* samples tested on the reference strains. Mean and standard deviation values are reported, showing statistically significant differences between Cell@ZIF-8 and Cell2@ZIF-8 composites (unpaired *t* test, * *p* < 0.05).

**Figure 7 jfb-14-00472-f007:**
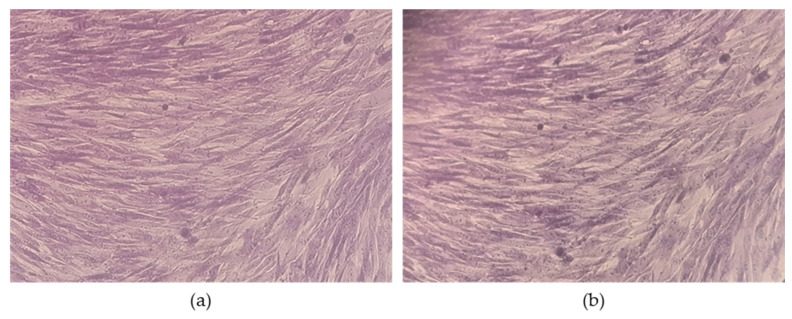
Light microscopy of Hel 299 stained with crystal violet (20 × original magnification). (**a**) Untreated cell control; (**b**) cells after the 24 h treatment with the *activated* Cell@ZIF-8 sample that does not alter cell morphology.

**Table 1 jfb-14-00472-t001:** Synthesis of cellulose/ZIF-8 composites.

Sample Name	Cellulose (g)	Zn(OAc)_2_∙2H_2_O(g) ^1^	2-HmIM (g) ^1^	Zn (wt %) ^2^	ZIF-8 (wt %) ^3^
Cell@ZIF-8 ^4^	2.50	2.30	8.60	9.1 ± 0.4	38.2 ± 0.4
Cell2@ZIF-8 ^5^	5.00	2.30	8.60	8.5 ± 0.4	30.2 ± 0.4

^1^ Zn(OAc)_2_∙2H_2_O 0.0759 M and 2-HmIM 0.759 M with respect to the final volume employed in the preparation (138 mL). ^2^ Measured using ICP analysis on the *activated* samples. The theoretical values are 14.0% and 9.3 wt % for Cell@ZIF-8 and Cell2@ZIF-8, respectively. ^3^ Based on the Zn wt % content. ^4^ Sheet size: 52 × 74 mm. ^5^ Sheet size: 105 × 148 mm.

**Table 2 jfb-14-00472-t002:** Surface area analysis of the cellulose/ZIF-8 composites.

Sample Name	ZIF-8 (wt %)	S_LAN_ (m^2^/g) ^1^	S_BET_ (m^2^/g) ^2^	V_TOTAL_ (cm^3^/g) ^3^
Cell@ZIF-8	38.2 ± 0.4	662 ± 1	660 ± 16	0.25
Cell2@ZIF-8	30.2 ± 0.4	455 ± 1	451 ± 15	0.18

^1^ Langmuir surface area. ^2^ surface area calculated using BET. ^3^ total pore volume.

**Table 3 jfb-14-00472-t003:** Antibacterial activity determined with the standardized disk-diffusion test. Data are expressed as ranges of the diameters of the inhibition zone in millimeters.

Sample Name	*S. epidermidis*	*P. aeruginosa*
Sterile paper disk	NA ^1^	NA
Cellulose bare	NA	NA
Cell@ZIF-8 *as-synthetized*	14−15	13−14
Cell@ZIF-8 *activated*	12−14	13 ± 16
Cell2@ZIF-8 *as-synthetized*	9−12	11−14
Cell2@ZIF-8 *activated*	9−13	10−14
GNT 10 µg	21−23	17−19
VNC 10 µg	14−16	NA

^1^ NA: Not appeared.

**Table 4 jfb-14-00472-t004:** Cell viability and LDH activity of Hel 299 cell line after a 48 h treatment with the samples. Data are mean percentages and standard deviations obtained in three biological replicates.

Sample Name	Cell Viability ^1^	LDH Activity ^1^
Sterile paper disk	100.0 ± 8.7	<5
Cellulose bare	99.0 ± 0.7	<5
Cell@ZIF-8 *as-synthetized*	0.20 ± 0.7	32.0 ± 4.7
Cell@ZIF-8 *activated*	58.1 ± 2.4	13.9 ± 2.5
Cell2@ZIF-8 *as-synthetized*	6.3 ± 1.3	29.4 ± 1.2
Cell2@ZIF-8 *activated*	84.3 ± 2.5	10.4 ± 2.4

^1^ Percentage values relative to the sterile paper disk.

**Table 5 jfb-14-00472-t005:** Antibacterial activity determined with the standardized disk-diffusion test. Data are expressed as ranges of the diameters of the inhibition zone in millimeters.

Clinical Isolate	Diameter of Inhibition	Antibiotic-Resistance Profile ^1,2^
*S. epidermidis* ATCC 12228	12−14	-
Strain 1 (MRSE) ^3^	11−13	CM^S^, DAP^S^, E^R^, GMN^S^, LVX^I^, OX^R^, TE^S^, SXT^S^, VA^S^
Strain 2 (MRSE) ^3^	13−15	CM^S^, DAP^S^, E^S^, GMN^S^, LVX^R^, OX^R^, TE^S^, SXT^S^, VA^S^
Strain 3	15−17	CM^S^, DAP^S^, E^S^, GMN^S^, LVX^S^, OX^S^, TE^S^, SXT^S^, VA^S^
Strain 4	18−20	CM^S^, DAP^S^, E^S^, GMN^S^, LVX^S^, OX^S^, TE^S^, SXT^S^, VA^S^
*P. aeruginosa* ATCC 27853	13−16	-
Strain 1	12−14	AN^S^, CAZ^S^, CAZ-AVI^S^, CT^S^, CIP^S^, FEP^s^, IMI^s^, MEM^S^, MEV^S^, TZP^S^
Strain 2	15−17	AN^S^, CAZ^S^, CAZ-AVI^S^, CT^S^, CIP^S^, FEP^S^, IMI^S^, MEM^S^, MEV^S^, TZP^S^
Strain 3 (MDR) ^4^	13−15	AN^S^, CAZ^I^, CAZ-AVI^S^, CT^S^, CIP^R^, FEP^S^, IMI^I^, MEM^S^, MEV^S^, TZP^S^
Strain 4 (MDR) ^4^	19−21	AN^S^, CAZ^R^, CAZ-AVI^S^, CT^S^, CIP^R^, FEP^R^, IMI^R^, MEM^I^, TZP^S^

^1^ AN = Amikacin; CAZ = Ceftazidime; CAZ-AVI = Ceftazidime/Avibactam; C/T = Ceftolozane/Tazobactam; CIP = Ciprofloxacin; CM = Clindamicyn; DAP = Daptomycin; E = Erythromycin; FEP = Cefepime; GMN = Gentamicin; IMI = Imipenem; LVX = Levofloxacin; MEM = Meropenem; MEV = Meropenem/Vaborbactam; OX = Oxacillin; SXT = Trimethoprim/Sulfamethoxazole; TZP = Piperacillin/Tazobactam; VA = Vancomycin. ^2^ R = Resistant; S = Susceptible; I = Intermediate, as defined following the EUCAST guidelines [38]. ^3^
*Staphylococcus* species resistant to oxacillin were declared, by convention, methicillin-resistant (MRSE). ^4^
*Pseudomonass* species defined MDR (multidrug) as resistant to at least three agents from a variety of antibiotic classes.

## Data Availability

The data presented in this study are available on request from the corresponding authors.

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
