# Peer review of "Cellulose/Zeolitic Imidazolate Framework (ZIF-8) Composites with Antibacterial Properties for the Management of Wound Infections"

_jfb, 2023, doi:10.3390/jfb14090472_

Round 1

Reviewer 1 Report

1. Pls remark the scale bar for the SEM.

2. Pls also provide the TEM, I want to check the core@shell nanostructure for the final products.

3. Some related work on the wound injections and ZIF-8 work could be cited such as J. Mater. Chem. B, 2023, 11, 6335–6345 and New J. Chem., 2022, 46, 13818–13837

4. The BET data should be collected for the full samples, and related ref could be cited, such as J. Mater. Chem. A, 2020, 8, 11933–11937.

5. From the Fig. 5, why the activated ZIF has loss its integrity?

6. Could you give the image for antibacterial activity?

work

Author Response

All changes in the text are highlighted.

Reviewer 2 Report

Manuscript No.: jfb-2539054-peer-review-v1

Title: Cellulose/zeolitic imidazolate framework (ZIF-8) composites with antibacterial properties for the management of wound infections

Journal of Functional Biomaterials

Reviewer's Decision: Accept after minor revision

The authors of this research work describe the polymeric composite system with antibacterial activities for healing and management. The research is significant and should be published in the Journal of Functional Biomaterials. However, the manuscript needs to be significantly improved before it can be published. As a result, I recommend accepting the manuscript after significant and satisfactory revisions. The following are the detailed comments:

1.     Title: Well written!

2.     Abstract: The abstract is a comprehensive summary of the whole research article. The abstract contains some language, grammatical, and formatting issues, and it is suggested to improve the grammar and English language problems. The abstract section is more introductive and contains methodology information. This information should be reduced by adding more result outcomes to strengthen your designed and evaluated research.

3.     Introduction: The introduction discusses guided tissue regeneration and potential biomaterial for fabricating composite polymeric membranes. The introduction and the rest of the manuscript have a few grammatical and formatting errors. Please improve the grammar, language, and formatting issues in the manuscript.

4.     References: The manuscript lacks the literature citation of some highly interesting, most recent relevant works; thus, the references are not current. Too many references and outdated citations may question the novelty of research work, and similar research has already been reported. In this regard, the author should refer to some of the most recent papers on hydrogel, such as

·       Stojanović, G. M. (2022). Multifunctional arabinoxylan-functionalized-graphene oxide-based composite hydrogel for skin tissue engineering. Frontiers in Bioengineering and Biotechnology10, 865059.

5.     Materials and methods: This section is missing, and please add it; otherwise, it may confuse the readers.

·       The author has used the same "degree" sign for temperature, contact angle, and angle. It is recommended to use the "degree" symbol accordingly throughout the manuscript.

6.     Results and Discussions: The following issue must be taken into consideration.

a.     The proper "minus" sign should be used rather than "dash" in the FTIR analysis and its discussion.

b.     The image quality of FTIR and XRD analysis must be revised.

7.     Conclusions: The conclusion section is the most important summary of a research article, and it should be based on the conclusion for the conclusion. The conclusion should be based on comparing the different used formulations by comparing the best result output.

8.     As per the comments given for the results and description.

In summary, the reported work has significant value; however, a major and thorough improvement/correction of language, grammar, syntax, etc., is necessary to improve the paper's quality and make it publishable in the Journal of Functional Biomaterials.

·       All the abbreviations should be defined before their 1st-time use.

English language and grammar is required to address carefully

Author Response

(The authors gave the same response as above.)

Reviewer 3 Report

The authors of the manuscript titled “Cellulose/zeolitic imidazolate framework (ZIF-8) composites with antibacterial properties for the management of wound infections” present an interesting study on the preparation of hybrid materials based on cellulose and zeolitic imidazolate framework 8 (ZIF-8) and the investigation of their potential antibacterial properties.

Below are some significant Comments that can be taken into consideration prior to publication.

1.     In the introduction part, the authors should mention other studies involving cellulose and ZIF-8. Are there any published works that investigate their potential antibacterial activity? The authors should also highlight the novelty of this work.

2.     The authors should define the abbreviation for HmIM.

3.     The authors should evaluate and describe the results more thoroughly.

4.     In Table 1 the authors present some data on the synthesized samples. How these experimental Zn (wt) values are compared with the theoretical values.

5.     The authors should include the characterization data of Cell2@ZIF8 in the manuscript (For example the data from SEM, XRD, AFM)

6.     In figure 3 and S3 the authors should identify the major peaks in the FTIR spectra.  Please evaluate more on FTIR data. Is there any difference in the FTIR spectra of Cell2@ZIF8 and Cell@ZIF8.

7.     The authors said that the sample Cell@ZIF-8 has an additional weight loss at 117 oC, but the TGA curve in Figure 4, shows this transition at higher temperature (around 160) and the decomposition of cellulose looks to be at higher temperature than 300 oC. Please explain.

8.     Can the authors calculate the inorganic component in samples Cell2@ZIF8 and Cell@ZIF8 from the TGA curves?

9.     The authors should define the peaks at the XRD diffractograms.

Author Response

(The authors gave the same response as above.)

Round 2

Reviewer 1 Report

accept. 

Reviewer 2 Report

All the comments have been addressed successfully and manuscript can be accepted in present form.

Some minor English issues are there and can be addressed during proofreading.